# In Vitro Anticancer Activity of Mucoadhesive Oral Films Loaded with *Usnea barbata* (L.) F. H. Wigg Dry Acetone Extract, with Potential Applications in Oral Squamous Cell Carcinoma Complementary Therapy

**DOI:** 10.3390/antiox11101934

**Published:** 2022-09-28

**Authors:** Violeta Popovici, Elena Matei, Georgeta Camelia Cozaru, Laura Bucur, Cerasela Elena Gîrd, Verginica Schröder, Emma Adriana Ozon, Adina Magdalena Musuc, Mirela Adriana Mitu, Irina Atkinson, Adriana Rusu, Simona Petrescu, Raul-Augustin Mitran, Mihai Anastasescu, Aureliana Caraiane, Dumitru Lupuliasa, Mariana Aschie, Victoria Badea

**Affiliations:** 1Department of Microbiology and Immunology, Faculty of Dental Medicine, Ovidius University of Constanta, 900684 Constanta, Romania; 2Center for Research and Development of the Morphological and Genetic Studies of Malignant Pathology, Ovidius University of Constanta, CEDMOG, 900591 Constanta, Romania; 3Clinical Service of Pathology, Sf. Apostol Andrei Emergency County Hospital, 900591 Constanta, Romania; 4Department of Pharmacognosy, Faculty of Pharmacy, Ovidius University of Constanta, 900001 Constanta, Romania; 5Department of Pharmacognosy, Phytochemistry, and Phytotherapy, Faculty of Pharmacy, Carol Davila University of Medicine and Pharmacy, 020956 Bucharest, Romania; 6Department of Cellular and Molecular Biology, Faculty of Pharmacy, Ovidius University of Constanta, 900001 Constanta, Romania; 7Department of Pharmaceutical Technology and Biopharmacy, Faculty of Pharmacy, Carol Davila University of Medicine and Pharmacy, 020956 Bucharest, Romania; 8Ilie Murgulescu” Institute of Physical Chemistry, Romanian Academy, 060021 Bucharest, Romania; 9Department of Oral Rehabilitation, Faculty of Dental Medicine, Ovidius University of Constanta, 900684 Constanta, Romania

**Keywords:** *Usnea barbata* (L.) F. H. Wigg dry acetone extract, oral squamous cell carcinoma, mucoadhesive oral films, usnic acid, CLS-354 cell line, blood cell cultures, oxidative stress, anticancer potential, antimicrobial activity

## Abstract

Oral squamous cell carcinoma (OSCC) is the most frequent oral malignancy, with a high death rate and an inadequate response to conventional chemotherapeutic drugs. Medical research explores plant extracts’ properties to obtain potential nanomaterial-based anticancer drugs. The present study aims to formulate, develop, and characterize mucoadhesive oral films loaded with *Usnea barbata* (L.) dry acetone extract (F-UBA) and to investigate their anticancer potential for possible use in oral cancer therapy. *U. barbata* dry acetone extract (UBA) was solubilized in ethanol: isopropanol mixture and loaded in a formulation containing hydroxypropyl methylcellulose (HPMC) K100 and polyethylene glycol 400 (PEG 400). The UBA influence on the F-UBA pharmaceutical characteristics was evidenced compared with the references, i.e., mucoadhesive oral films containing suitable excipients but no active ingredient loaded. Both films were subjected to a complex analysis using standard methods to evaluate their suitability for topical administration on the oral mucosa. Physico-chemical and structural characterization was achieved by Fourier transform infrared spectroscopy (FTIR), X-ray diffraction (XRD), thermogravimetric analysis (TGA), scanning electron microscopy (SEM), and atomic force microscopy (AFM). Pharmacotechnical evaluation (consisting of the measurement of specific parameters: weight uniformity, thickness, folding endurance, tensile strength, elongation, moisture content, pH, disintegration time, swelling rate, and *ex vivo* mucoadhesion time) proved that F-UBAs are suitable for oral mucosal administration. The brine shrimp lethality (BSL) assay was the F-UBA cytotoxicity prescreen. Cellular oxidative stress, caspase 3/7 activity, nuclear condensation, lysosomal activity, and DNA synthesis induced by F-UBA in blood cell cultures and oral epithelial squamous cell carcinoma (CLS-354) cell line were investigated through complex flow cytometry analyses. Moreover, F-UBA influence on both cell type division and proliferation was determined. Finally, using the resazurin-based 96-well plate microdilution method, the F-UBA antimicrobial potential was explored against *Staphylococcus aureus* ATCC 25923, *Pseudomonas aeruginosa* ATCC 27353, *Candida albicans* ATCC 10231, and *Candida parapsilosis* ATCC 22019. The results revealed that each UBA-loaded film contains 175 µg dry extract with a usnic acid (UA) content of 42.32 µg. F-UBAs are very thin (0.060 ± 0.002 mm), report a neutral pH (7.01 ± 0.01), a disintegration time of 146 ± 5.09 s, and an ex vivo mucoadhesion time of 85 ± 2.33 min, and they show a swelling ratio after 6 h of 211 ± 4.31%. They are suitable for topical administration on the oral mucosa. Like UA, they act on CLS-354 tumor cells, considerably increasing cellular oxidative stress, nuclear condensation, and autophagy and inducing cell cycle arrest in G0/G1. The F-UBAs inhibited the bacterial and fungal strains in a dose-dependent manner; they showed similar effects on both *Candida* sp. and higher inhibitory activity against *P. aeruginosa* than *S. aureus*. All these properties lead to considering the UBA-loaded mucoadhesive oral films suitable for potential application as a complementary therapy in OSCC.

## 1. Introduction

Approximately 3–10% of all cancer mortality is known to be contributed by oral cancer [1]. Oral cancer incidence increases due to environmental conditions and harmful habits of the modern lifestyle: pollution, diet and nutrition, tobacco smoking [2], betel quid chewing [3], and alcohol consumption [4,5]. These factors, associated with a hereditary predisposition, chronic inflammation, and infectious diseases, have contributed to the increased risk of developing oral cavity malignancies [6]. Oral squamous cell carcinoma (OSCC) represents 90% of oral neoplasia [7]; it is the sixth most common cancer in the world [8] and has an overall 5-year survival rate below 50% [9] due to its modest outcomes, tardive diagnosis, and inadequate response to chemoradiation therapy [10]. Furthermore, after the current treatment protocol [10], the quality of life of patients with oral cancer is substantially diminished [11], and restrictions on food intake could lead to malnutrition [12].

Consequently, global medical research focuses on discovering and developing alternative therapies against cancer, investigating the properties of plant extracts to obtain potential nanomaterial-based anticancer drugs. According to Dehelean et al. [13], the anticancer potential of natural products could be expressed as chemotherapeutic effects (due to their innate antitumor activity), chemopreventive action (maintaining a low risk of developing cancer or keeping it from coming back), and sensitizers in multidrug resistance.

In a recent comprehensive review, Prakash et al. [14] described various medicinal plants with beneficial effects against oral cancer (*Ocimum sanctum* L., *Curcuma longa* L., *Vaccinium corymbosum* L., *Vaccinium macrocarpon* Aiton, *Momordica charantia* L., *Azadirachta indica* A. Juss, *Senegalia Catechu* (L.f.) P.J.H. Hurter & Mabb., *Dracaena cinnabari* Balf.f., *Piper nigrum* L., and *Zingiber officinale* Roscoe) and their active secondary metabolites (curcumin, nimbolide, resveratrol, anthocyanins, piperine, and eugenol). Their list could be completed with *Melissa officinalis* L. [15], *Gracilaria tenuistipitata* C.F. Chang & B.M. Xia [16], *Cynara cardunculus* L. [17], *Imperata cylindrica* (L.) P.Beauv. [18], *Caesalpinia sappan* L. [19], *Angelica gigas* Nakai [20], *Aloe barbadensis* Miller [21], *Padina gymnospora* [22] and *Usnea barbata* (L.) F.H.Wigg [23,24,25]. The principal mechanism triggered in the OSCC tumor cell line is oxidative stress, which induces DNA damage and cell cycle arrest, leading to apoptotic cell death [16,17,23,24]. Moreover, these medicinal plants could also be helpful in oral cancer chemoprevention due to other associated bioactivities (such as antimicrobial, anti-inflammatory, antioxidant, and wound-healing effects).

Numerous studies described plant-based oral formulations tested for their efficacy in improving oral hygiene, salivary microbial flora, and gingival health. Pérez Zamora et al. developed herbal buccal films with in vitro antibacterial and anti-inflammatory effects containing *Lippia turbinata* Griseb. and *Lippia alba* (Mill.) Britton & P.Wilson extracts [26]. Pagano et al. [27] incorporated grape seed extract in bioadhesive patches with a combination of acacia gum/PVP/cyclic dextrin, aiming for a wound dressing effect. Utama-ang et al. [28] developed Thai rice films loaded with ginger extract for oral antimicrobial properties.

Other authors formulated fast-dissolving herbal films with *Eclipta prostrata* (L.) leaves extract for mouth ulcers [29] and oral patches containing *Glycyrrhiza* complex herbal extract [30], and *Myrtus communis* L. (Myrtle) [31] for aphthous stomatitis. Nam et al. [32] proposed *Angelica gigas* Nakai extract-loaded fast-dissolving nanofiber based on poly(vinyl alcohol) and Soluplus for oral cancer therapy. Recently, Chiaoprakobkij et al. [33] examined in vitro anticancer and antibacterial effects of *Garcinia mangostana* L. extract incorporated in bacterial cellulose-gelatin films.

The present study aimed to develop and characterize mucoadhesive oral films based on HPMC and PEG 400 containing *U. barbata* dry acetone extract. Moreover, our work investigated in vivo cytotoxicity on an animal model and in vitro antitumor and antimicrobial activities of F-UBA for potential application in oral cancer prevention and therapy.

## 2. Materials and Methods

### 2.1. Materials

All chemicals, reagents, and standards used in this study were of analytical grade. Polyethylene glycol 400 (PEG 400), hydroxypropyl methylcellulose (HPMC), usnic acid standard 98.1% purity, propidium iodide (PI) 1.0 mg/mL, dimethyl sulfoxide (DMSO), and antibiotics mix solution 100 µL/mL with 10 mg streptomycin, 10,000 U penicillin, 25 µg amphotericin B per 1 mL were provided by Sigma-Aldrich Chemie GmbH (Taufkirchen, Germany). Annexin V Apoptosis Detection Kit and flow cytometry staining buffer (FCB) were purchased from eBioscience^TM^ (Frankfurt am Main, Germany) and RNase A 4 mg/mL from Promega (Madison, WI, USA). Magic Red^®^ Caspase-3/7 Assay Kit, Reactive Oxygen Species (ROS) Detection Assay Kit, and EdU i-Fluor 488 Kit were supplied by Abcam (Cambridge, UK).

The OSCC cell line (CLS-354) and the culture medium Dulbecco’s Modified Eagle’s Medium (DMEM) High Glucose basic supplemented with 4.5 g/l glucose and L-glutamine and 10% fetal bovine serum (FBS) were provided by CLS Cell Lines Service GmbH (Eppelheim, Germany). Trypsin-ethylenediamine tetra acetic acid (Trypsin EDTA) and the media for blood cells Dulbecco’s phosphate-buffered saline with MgCl_2_ and CaCl_2_, FBS, and L-Glutamine (200 mM) solution were purchased from Gibco^TM^ Inc (Billings, MT, USA).

The blood samples were collected from a non-smoker healthy donor (B Rh+ blood type), according to Ovidius University of Constanta Ethical approval code 7080/10 June 2021 and Donor Consent code 39/30 June 2021.

*Artemia salina* eggs and Artemia salt (Dohse Aquaristik GmbH & Co. Gelsdorf, Germany) were purchased online from https://www.aquaristikshop.com/ (accessed on 5 May 2022).

The microbial cell lines (*S. aureus* ATCC 25923, *P. aeruginosa* ATCC 27353, *C. albicans* ATCC 10231, and *C. parapsilosis* ATCC 22019) were obtained from the Microbiology Department, S.C. Synevo Romania SRL, Constanta Laboratory, under partnership agreement No 1060/25 January 2018 with the Faculty of Pharmacy, Ovidius University of Constanta. Culture medium Mueller-Hinton agar (MHA) was supplied by Thermo Fisher Scientific, GmbH, Dreieich, Germany; RPMI 1640 Medium and Resazurin solution (from In Vitro Toxicology Assay Kit, TOX8-1KT, Resazurin based) were purchased from Sigma-Aldrich Chemie GmbH (Taufkirchen, Germany).

*U. barbata* lichen was harvested in March 2021 from the forest localized in the Călimani Mountains (47 °29′ N, 25 °12′ E, and 900 m altitude). It was identified by the Department of Pharmaceutical Botany of the Faculty of Pharmacy, Ovidius University of Constanta, using standard methods. A voucher specimen is maintained in the Herbarium of Pharmacognosy Department, Faculty of Pharmacy, Ovidius University of Constanta (Popovici 3/2021, Ph-UOC).

### 2.2. Lichen Extract

The dried lichen was ground in an LM 120 laboratory mill (PerkinElmer, Waltham, MA, USA) and passed through the no. 5 sieve [34]. The obtained moderately fine lichen powder (particle size ≤ 315 μm) was subjected to Soxhlet extraction in acetone at 55–60 °C for 8 h, followed by solvent evaporation; the method is described in a previously published study [34]. The *U. barbata* dry acetone extract (UBA) was preserved in a freezer (Sirge^®^ Elettrodomestici—SAC Rappresentanze, Torino, Avigliana, Italy) at −18 °C until used for mucoadhesive oral films preparation [34].

### 2.3. Formulation and Manufacturing of Mucoadhesive Oral Films

Because the UBA-loaded mucoadhesive films are designated for oral administration and residual amounts of acetone may be found in the final pharmaceutical product, it was decided to select a more appropriate solvent for *U. Barbata* dry acetone extract. Thus, a mix of 1:1 (*w*/*w*) ethyl alcohol and isopropyl alcohol was used for UBA solubilization. The solubility of UBA in the previously mentioned combination was 2.5% (*w*/*w*); thus, UBA dosage in the developed films was established. A Mettler Toledo AT261 balance (with 0.01 mg sensitivity) was used to weigh the ingredients.

HPMC K100 [35] with a viscosity of 100 mPa was chosen for the film matrix formation, considering its high hydrophilic character, stability, and biocompatibility [36,37,38]. PEG 400 was used as a plasticizer because it offers excellent flexibility to the polymer base [39,40].

Without UBA, the references (R) were prepared to investigate the active ingredient’s influence on the mucoadhesive oral films’ physicochemical and pharmacotechnical characteristics and evaluate the extract’s efficacy. HPMC dispersion was prepared by adding the weighted polymer to water at room temperature and stirring for 30 min at 700 rpm, using a Heidolph MR 3001K magnetic stirrer (Schwabach, Germany). PEG 400 was incorporated into the polymer matrix. The lichen extract solution was obtained by dissolving UBA in the ethanol: isopropanol mixture and then poured over the polymer dispersion under continuous stirring in the same conditions. The formed gel systems were left overnight for deaeration in ambient conditions. The dispersions were placed in a thin layer on Petri glass plates and dried for 24 h at room temperature. Next, the dried films were detached from the glass plate surface and cut into patches of 1.5 × 2 cm sizes.

### 2.4. Physico-Chemical Characterization of the UBA-Loaded Mucoadhesive Oral Films

#### 2.4.1. Fourier Transform Infrared Spectroscopy

Fourier transform infrared spectroscopy (FTIR) measurements were performed using a NICOLET 6700 FTIR Spectrometer (Grayslake, IL, USA) with a Smart DuraSamplIR HATR (Horizontal Attenuated Total Reflectance) accessory and a laminated–diamond crystal (Thermo Electron Corporation, Waltham, MA, USA). The FTIR spectra were assessed in the range of 4000–400 cm^−1^ using a DTGS KBr detector at a resolution of 4 cm^−1^. All spectra were plotted in transmittance mode.

#### 2.4.2. Powder X-ray Diffractometry

Powder X-ray diffraction (XRD) measurements were recorded using a Rigaku Ultima IV diffractometer (Rigaku Corporation, Tokyo, Japan) in parallel beam geometry with a step size of 0.02° and a speed of 2°/min over a range of 5–60°. A CuKα tube (λ = 1.54056 Å) operating at 40 kV and 30 mA was the source of the X-rays.

#### 2.4.3. TG/DTA Measurements

The conditions of both analyses—thermogravimetric analysis (TGA) coupled with differential thermal analyses (DTA)—comprised a synthetic air atmosphere with a heating rate of 10 °C min^−1^ and a flow rate of 80 mL/ min. Both films’ thermogravimetric curves were carried out on a Mettler Toledo TGA/SDTA 851^e^ thermogravimetric analyzer (Mettler-Toledo GmbH, Greifensee, Switzerland).

#### 2.4.4. Scanning Electron Microscopy (SEM)

Morphological analysis was made on a high-resolution scanning electron microscope FEI Quanta3D FEG (Thermo Fisher Scientific, GmbH, Dreieich, Germany).

#### 2.4.5. Atomic Force Microscopy (AFM)

AFM analysis was made using an XE-100 microscope from Park Systems (Suwon-si, South Korea) in non-contact mode. Sharp tips, NCHR from NanosensorsTM, with typically ~8 nm radius of curvature, ~125 mm mean length, 30 mm mean width, ~42 N/m force constant, and ~330 kHz resonance frequency was used. The image processing and roughness evaluation were performed using the XEI program (v 1.8.0) from Park Systems. The detailed surface profile of the scanned films was presented in “enhanced contrast” view mode.

### 2.5. Pharmacotechnical Analysis of the UBA-Loaded Mucoadhesive Oral Films

#### 2.5.1. Weight Uniformity

The weight uniformity was achieved on 20 films of each type (F-UBA and R) that were individually weighed, and the average was calculated.

#### 2.5.2. Thickness

This parameter was assessed with a Yato Trading CO., Ltd., Shanghai, China, digital micrometer with a 0–25 mm measurement range and 0.001 mm resolution on 20 films of each formulation. The mean and the standard deviation were calculated.

#### 2.5.3. Folding Endurance

To test folding endurance, 10 films of each formulation were folded and rolled repeatedly, at the same place, until they broke, or up to 300 times [41]. The folding times were recorded and reported as folding endurance values.

#### 2.5.4. Tensile Strength and Elongation Ability

Both parameters were determined using an LR 10K Plus digital tensile force tester for universal materials (Lloyd Instruments Ltd., West Sussex, UK). For the test, 5 films of each type (F-UBA and R) were placed in a vertical position between apparatus 2 braces, and then the test was performed at a speed of 30 mm/min from a distance of 30 mm.

The tensile strength (kg/mm^2^) and elongation at break (%) were calculated using the following equations:(1)$$\mathrm{Tensile}\;\mathrm{strength}\left(\mathrm{kg}/{\mathrm{mm}}^{2}\right)=\frac{Force\;at\;breakage\;\left(\mathrm{kg}\right)}{Film\;thickness\;\left(\mathrm{mm}\right)\times Film\;width\;\left(\mathrm{mm}\right)}$$
(2)$$Elongation\;\%\;=\frac{Increase\;in\;film\;length\;\left(\mathrm{cm}\right)}{Inital\;film\;length\;\left(\mathrm{cm}\right)}\times 100$$

#### 2.5.5. Moisture Content

The moisture content was evaluated as the loss on drying using a thermogravimetric method with an HR 73 Mettler Toledo halogen humidity analyzer (Mettler-Toledo GmbH, Greifensee, Switzerland) [42]. Five films of each formulation (F-UBA and R) were tested.

#### 2.5.6. Surface pH Value

For pH testing, 5 films of each formulation (F-UBA and R) were moistened for 5 min at room temperature with 1 mL distilled water (pH 6.5 ± 0.5). The pH was registered by touching the film surface with the electrode of a CONSORT P601 pH-meter (Consort bvba, Turnhout, Belgium).

#### 2.5.7. In Vitro Disintegration Time

This parameter was assessed on 6 films of each formulation (F-UBA and R) in simulated saliva phosphate buffer pH 6.8 at 37 ± 2 °C [43], using an Erweka DT 3 apparatus (Erweka^®^ GmbH, Langen, Germany). The time required for both films’ total disintegration was registered.

#### 2.5.8. Swelling Ratio

The films were placed in Petri plates on a 1.5% agar gel and incubated at 37 ± 1 °C. They were weighed every 30 min for 6 h, and the swelling ratio was calculated according to the following equation:(3)$$Swelling\;ratio=\frac{Wt-Wi}{Wi}\times 100$$
where *Wt* is the film weight at time *t* after the incubation and *Wi* is the initial weight [44,45,46].

#### 2.5.9. Ex Vivo Mucoadhesion Time

The method used to determine the residence time on a detached porcine oral mucosa was adapted to those described by Gupta et al. [47]. The fat layer and any tissue residue were removed from the membrane surface, then rinsed with distilled water and a phosphate buffer pH of 6.8 at 37 °C and fixed on a glass plate. Each film was hydrated in the center with 15 μL phosphate buffer and pressed on the membrane surface for 30 s. The glass plate was in a 200 mL phosphate buffer of pH 6.8 and kept at 37 °C for 2 min. The paddle with a stirring rate of 28 rpm was operated to reproduce the oral cavity conditions. The bioadhesion time was established by measuring the necessary time for each film to detach from the oral mucosa. All tests were performed in triplicate.

### 2.6. Evaluation of the Cytotoxic Activity of UBA-Loaded Mucoadhesive Oral Films on A. salina Larvae

#### 2.6.1. Sample Preparation

The F-UBA was placed in a diluted buffer (1 mL) and incubated for 15 min at 37 °C; then, its homogenous dispersion in the buffer solution was observed.

#### 2.6.2. BSL Assay

*Artemia salina* (brine shrimp) was used as an animal model for the UBA-loaded mucoadhesive oral films’ cytotoxicity investigation, adapted from Nazir et al. [48]. The *A. salina* larvae were obtained under continuous light and aeration conditions, at a temperature of 20 °C, by introducing the cysts of *A. salina* for 24–48 h in a saline solution of 0.35%. The brine shrimp larvae in the first stage (instar I) were introduced in 0.3% saline solution into experimental pots (with a volume of 1 mL) [49]. The analysis was compared with a blank (untreated nauplii) to obtain accurate results regarding the F-UBA cytotoxic effects. The nauplii were not fed during the test to not interfere with the tested extracts. Their evolution was investigated after 24 h and 48 h; the larvae had embryonic energy reserves as lipids throughout this period.

Morphological changes of brine shrimp larvae [50,51] after 24 and 48 h were observed at VWR microscope VisiScope 300D (VWR International, Radnor, PA, USA).

#### 2.6.3. Fluorescent Microscopy

After 48 h, the brine shrimp larvae were stained with 3% acridine orange (Merck Millipore, Burlington, MA, USA) for 5 min. The samples were subjected to drying for 15 min in darkness and placed on the microscope slides.

#### 2.6.4. Data Processing

The microscopic images were achieved using a VWR microscope VisiScope 300D (VWR International, Radnor, PA, USA) with a Visicam X3 camera (VWR International Radnor, PA, USA) at 100× and 400× magnifications and processed with VisiCam Image Analyzer 2.13.

Using an OPTIKA B-350 microscope (Ponteranica, BG, Italy) blue filter (λex = 450–490 nm; λem = 515–520 nm) and green filter (λex = 510–550 nm; λem = 590 nm), fluorescent microscopy images were obtained [52]. The FM images at 200× and 400× magnification were processed with Optikam Pro 3 Software (OPTIKA S.R.L., Ponteranica, BG, Italy).

All observations were performed in triplicate.

### 2.7. In Vitro Analysis of the Effects of UBA-Loaded Mucoadhesive Oral Films on Human Normal Blood Cells and OSCC CLS-354 Cell Line

#### 2.7.1. Equipment

The Attune Acoustic focusing cytometer (Applied Biosystems, Bedford, MA, USA) was the platform for the in vitro cytotoxicity analysis of UBA-loaded mucoadhesive oral films. Before cell analysis, the flow cytometer was first set by using fluorescent beads Attune performance tracking beads, labeling, and detection (Life Technologies, Europe BV, Bleiswijk, Netherlands) [53], with a standard size (four intensity levels of beads population). The cell amount was established by enumerating cells below 1 µm [54]. Using forward scatter (FSC) and side scatter (SSC), more than 10,000 cells per sample for each analysis were gated.

#### 2.7.2. Data Processing

Flow cytometry data were achieved using Attune Cytometric Software v.1.2.5, Applied Biosystems 2010 (Bedford, MA, USA).

#### 2.7.3. Human Blood Cells Cultures

The blood sample was collected into heparin vacutainers. Then, the heparinized blood (1.0 mL) was added to untreated Nunclon Vita Cell culture 6-well plates (Kisker Biotech GmbH & Co.KG, Steinfurt, Germany), together with 6.0 mL of Dulbecco’s phosphate-buffered saline with MgCl_2_ and CaCl_2_ medium supplemented with 10% bovine fetal serum, L-glutamine, and antibiotics mix solution. They were incubated in a Steri-Cycle™ i160 CO_2_ Incubator (Thermo Fisher Scientific Inc., Waltham, MA, USA) with 5% CO_2_ at 37 °C. After 72 h, the blood cell cultures were treated with the samples and controls. Then, the cells were subjected to 24 h of incubation in the same conditions [53]. All flow cytometry analyses were performed after this incubation time.

#### 2.7.4. CLS-354 Cell Line, Cells Culture

The CLS-354 tumor cells were cultured in DMEM high glucose supplemented with antibiotic mix solution in humidity conditions of 5% CO_2_ at 37 °C for 7 days [55]. Then, the cells were dissociated with trypsin-EDTA, centrifugated at 3000 rpm for 10 min in a Fisher Scientific GT1 centrifuge (Thermo Fisher Scientific Inc., Waltham, MA, USA), and distributed in Millicell™ 24-Well Cell Culture Microplates (Thermo Fisher Scientific Inc., Waltham, MA, USA). After treatment, the cells were incubated for 24 ore in the same conditions [25]. All the flow-cytometry analyzes were performed after this incubation period.

#### 2.7.5. Samples and Control Solutions

The UBA-loaded mucoadhesive oral films were dissolved in the suitable culture medium for both types of cells, with 1% DMSO. As a positive control, usnic acid (125 µg/mL in 1% DMSO) was selected, and the negative control was 1% DMSO.

### 2.8. Evaluation of Total ROS Activity

ROS assay stain solution (100 µL) was added to each 1 mL of cell culture in flow cytometry tubes and well-mixed. Then, the cells were incubated in a 5% CO_2_ atmosphere at 37 °C for 60 min. After this process, the cells were analyzed by flow cytometry, using a 488 nm excitation and green emission for ROS (BL1 channel).

### 2.9. Evaluation of Caspase 3/7 Activity

Both cell cultures (300 µL) were transferred in flow cytometry tubes; then, 20 µL of a Magic Red^®^ Caspase-3/7 Substrate—MR-(DEVD)2-solution—was added and well-mixed with the cells. Next, 20 µL of PI was added. After incubation, 1 mL FCB was added. Then, the early stages of cell apoptosis by activating caspases 3/7 (DEVD-ases) [56] were analyzed through flow cytometry using a 488 nm excitation, red emission for MR-(DEVD)2—BL3 channel, and orange emission for PI—BL2 channel.

### 2.10. Evaluation of Nuclear Condensation and Lysosomal Activity

Magic Red^®^ Caspase-3/7 Assay Kit contains Hoechst 33,342 stain (200 μg/mL) and acridine orange (AO, 1.0 µM). Hoechst 33,342 is a cell-permeant nuclear stain [57]; when it is linked to double-chain DNA, it emits blue fluorescence, highlighting condensed nuclei in apoptotic cells. Acridine orange is a chelating dye that can be used to reveal lysosomal activity [58]. Here, 300 µL of each cell culture was introduced in flow cytometry tubes; then, 2 µL of Hoechst 33,342 stain was added, and the cells were mixed well [53]. After these operations, 50 µL of 1.0 µM AO was added; the cells were incubated at room temperature into darkness for 30 min. After incubation, 1 mL FCB was added; the cells were examined at the flow cytometer under the following conditions: an excitation of 488 nm, the UV excitation, and blue emission for Hoechst 33,342 (VL2), and green emission acridine orange (BL1 channel) [53]. 

### 2.11. Cell Cycle Analysis

A cell culture volume of 1 mL was washed in FCB, introduced in flow cytometry tubes, and fixed with 50 µL ethanol for 30 min [53]. Next, the cells were treated with PI (20 µg/mL) and RNase A (30 µg/mL) and incubated at room temperature, into darkness, for 30 min [53]. Then, 1 mL FCB was added, and the cell cycle distribution was detected at the flow cytometer in the following conditions: a 488 nm excitation and orange emission for PI (BL2 channel) [53].

### 2.12. Annexin V-FITC Apoptosis Assay

The blood cells and CLS-354 tumor cells were incubated in flow cytometry tubes with 2 µL Annexin V-FITC and 2 µL PI (20 µg/mL) for 30 min, at room temperature, in darkness. After incubation, 1 mL of FCB was added. All viable cells, early apoptotic cells, late apoptotic cells, and necrotic cells were examined at a flow cytometer using the following conditions: an excitation of 488 nm and two emission types: green for Annexin V-FITC (BL1 channel) and orange for PI (BL2 channel) [53].

### 2.13. Evaluation of Cell Proliferation

Volumes of 1 mL of both cell cultures were incubated with 50 µM EdU (500 µL) at 37 °C for 2 h. Then, both cell types were fixed with 4% paraformaldehyde in PBS (100 µL) and permeabilized with Triton X-100 (100 µL). After washing in 3% buffer sodium azide (BSA) and centrifuging at 300 rpm for 5 min, at 4 °C, the cells were incubated with a reaction mix (500 µL) for 30 min at room temperature into darkness. Then, they were washed in permeabilization buffer and centrifuged at 300× *g* rpm for 5 min, at 4 °C. After these procedures, 1 mL FCB was added, and the cells were examined by flow cytometry, using a 488 nm excitation and green emission for EdU-iFluor 488 (BL1).

### 2.14. Antimicrobial Activity Evaluation by Resazurin-Based 96-Well Plate Microdilution Method

#### 2.14.1. Inoculum Preparation

The direct colony suspension method (CLSI) was used for preparing the bacterial inoculum. First, bacterial colonies selected from a 24 h agar plate were suspended in an MHA medium. The bacterial inoculum was accorded to the 0.5 McFarland standard, measured at Densimat Densitometer (Biomerieux, Marcy-l’Étoile, France) with around 10^8^ CFU/mL (CFU = colony-forming unit). The fungal inoculum was prepared using the same method, adjusting the RPMI 1640 with fungal colonies to the 1.0 McFarland standard, with 10^6^ CFU/mL.

#### 2.14.2. Samples and Standards

The F-UBA was dissolved in 1 mL of diluted phosphate buffer. As standards, Ceftriaxone (Cefort 1 g Antibiotice SA Iasi, Romania) solutions 30 mg/mL and 122 mg/mL in distilled water were used for bacteria. Cefort powder was weighted at Partner Analytical balance (Fink & Partner GmbH, Goch, Germany) and dissolved in distilled water. Terbinafine solution 10.1 mg/mL (Rompharm Company S.R.L., Otopeni, România) was selected as standard for *Candida* sp.

#### 2.14.3. Resazurin-Based 96-Well Plate Microdilution Method

All successive steps were performed in an Aslair Vertical 700, laminar flow, microbiological protection cabinet (Asal Srl, Cernusco (MI) Italy). In four 96-well plates, we performed seven serial dilutions, adapting the protocol described by Fathi et al. [59] and Elshikh et al. [60].

The 96-well plates were incubated for 24 h at 37 °C for bacteria and 35 °C for yeasts in a My Temp mini Z763322 Digital Incubator (Benchmark Scientific Inc., Sayreville, NJ, USA).

#### 2.14.4. Reading and Interpreting

After 24 h incubation, the colors from 96-well plates were examined to see the differences between standard and samples [61]. Moreover, they were evaluated at the Smart LED Illuminator (Kaneka Eurogentec S.A., Seraing, Belgium) at the wavelength of 470 nm, and the active sample concentrations were compared with the Standard antibiotic ones. The F-UBA microdilutions whose color was similar to standard antibiotic ones were highlighted. For yeasts, the color chart of the resazurin dye reduction method was used [62,63].

### 2.15. Data Analysis

All analyses were performed in triplicate, and the results were presented as means ± standard deviations (SD). They are expressed as percent (%) of the cells for apoptosis, caspase 3/7 activity, nuclear condensation, autophagy, cell cycle, DNA synthesis, and count (×10^4^) of ROS for cellular oxidative stress. The flow cytometry data were collected with SPSS v.23, 2015 (IBM, Armonk, NY, USA). The Levene test was analyzed for the homogeneity of sample variances. The paired *t*-test was used (*p* < 0.05 was considered statistically significant) to establish the differences between samples and controls. The principal component analysis (PCA) was performed using XLSTAT 2022.2.1. by Addinsoft (New York, NY, USA) [34].

## 3. Results

### 3.1. Physico-Chemical Characterization of the UBA-Loaded Mucoadhesive Oral Films

The composition of R and F-UBA is illustrated in Table 1.

Both formulations (F-UBA and reference) led to thin, non-sticky, transparent, and homogenous mucoadhesive with smooth and uniform surfaces; no cracks or air bubbles were visible. UBA-loaded mucoadhesive oral films have a yellowish-greenish-faint brown color (due to UBA), and R ones are colorless (Appendix A from Appendix A). The final F-UBA has 175 µg UBA, with 42.32 µg usnic acid, from a total phenolic content of 150.997 µg. 

#### 3.1.1. FTIR Spectra

The FTIR spectra of reference (R) and UBA-loaded mucoadhesive oral films are presented in Figure 1a.

The broad absorbance peak at 3443 cm^−1^ corresponds to the O-H stretching vibration of HPMC [64]. The absorption peak at 2915 cm^−1^ is assigned to the stretching vibration of the C–H band, and the peaks at 1455 cm^−1^ correspond to the deformation vibration of C–H_2_ [65]. The absorption peaks at 1118 cm^−1^ and 940 cm^−1^ are attributed to the C–O–C asymmetric stretching vibrations and β-glycosidic linkages, respectively [66,67].

Instead, the FTIR spectrum of F-UBA (Figure 1a, magenta line) shows the presence of the same peaks found in the reference (Figure 1a, black line), with a slight displacement and lower intensities. These observations proved the dispersion of UBA in the polymer matrix.

#### 3.1.2. XRD Analysis

The X-ray diffraction patterns of reference and F-UBA are shown in Figure 1b. The diffraction patterns obtained for the UBA-loaded polymer matrix (F-UBA) are similar to that of the reference, with a change in intensity. The reference shows the diffraction pattern of HPMC film with two vast diffraction peaks at 7.94° and 20.55°, which indicate its semi-crystalline nature [68,69].

These peaks are characteristic of planes (101) and (002). The UBA incorporation does not affect the parent HPMC matrix.

#### 3.1.3. TG/DTA Measurements

The thermogravimetric method was performed in a linear heating program under an air atmosphere to study the films’ thermal behavior. Thermal analysis of the films shows several steps of mass loss (Table 1 and Figure 1c).

The first step, similar for both films (reference and F-UBA), occurs up to 100 °C and is mainly due to the dehydration process (the used solvent and physisorbed water). The reference lost about 1.2%, while F-UBA lost about 2.5%.

In the second step, R lost about 86.9% of the mass, while F-UBA lost about 85.5%. The decomposition of the organic compounds occurs between 200 and 400 °C (in the films, the thermal degradation of HPMC polymer) [70].

The third step, between 400 and 500 °C, is due to the thermal decomposition of the other more stable organic groups.

The thermodynamic parameters (mass losses and maximum decomposition temperatures obtained from DTA curves) associated with the three decomposition steps are registered in Table 1.

#### 3.1.4. SEM Analysis

SEM images of reference (R) and UBA-loaded mucoadhesive oral films are shown in Figure 2a,b. The reference (Figure 2a) shows a smooth surface morphology with some small round particles. When UBA was added to the reference, a difference could be observed in the F-UBA’s morphology. Compared with the reference, a rougher surface was noted for the mucoadhesive film with UBA incorporated (Figure 2b).

#### 3.1.5. AFM Measurements

The reference (Figure 2c) exhibits a surface with smaller features, particle-like, as can be observed mainly in the lower part of Figure 2c, in the dashed circle surrounding such an area of particles. Nevertheless, some large pits (surface cavities) are observed on the reference’s surface. The arbitrary lines plotted below the AFM images of the R film suggest a similar Z-scale (oscillations level) of about 120 nm. (Figure 2e)

Figure 2d shows the UBA-loaded mucoadhesive oral film. Its morphology is less uniform than the reference, exhibiting small protuberances. F-UBA roughness and peak-to-valley parameters increase compared with R (Figure 2f).

Calculating the corrugation parameters using the whole scanned area, i.e., 8 × 8 µm^2^, highlights the rougher morphology of F-UBA (Figure 2h). Using the scale of 3 × 3 µm^2^ (Figure 2g,h), it was observed that the roughness shows the same trend for the small areas: from the morphological point of view, the films are relatively uniform. The AFM images accord with the SEM analysis regarding both films’ morphology.

### 3.2. Pharmacotechnical Characterization of UBA-Loaded Mucoadhesive Oral Films

The pharmacotechnical properties of both films—F-UBA and reference (R)—are displayed in Table 1. 

Reference has around 66 mg, while F-UBA shows an average weight of 70 mg, containing 175 mg UBA with a usnic acid content of 42.32 µg.

Both films are very thin, between 0.058 and 0.060 mm.

The folding endurance values were above 300 for both types of films.

F-UBA has a tensile strength of 3.02 kg/mm^2^ and a 47.26% elongation, while the R values were 2.88 kg/mm^2^ and 49.25%, respectively.

F-UBA has a moisture content of 4.11%, while R presents a 3.98% loss on drying. 

Both films present a neutral pH (7.01 for F-UBA and 7.04 for R), indicating good biocompatibility with the buccal mucosa and high tolerability.

The F-UBA displayed an in vitro disintegration time in a simulated saliva medium of 146 s, compared to R one of 138 s.

All data obtained reported no significant differences between F-UBA and R (*p* > 0.05).

The swelling ratio registered for mucoadhesive oral films (F-UBA and R) over the 6 h is shown in Figure 1d.

The swelling behavior of the two films (F-UBA and R) was similar, with a higher swelling rate in the first 150 min. The hydration process continued with a much slower profile, with erosion occurring after 6 h. The swelling index of F-UBA was 211% after 6 h, and that of R, 204% (Table 1, Figure 1d)

Both mucoadhesive films exhibited good retention times on the oral mucosa: 85 min for F-UBA and 82 min for R; it could be observed that UBA incorporation slightly increased the bioadhesion time by 3 min (Table 1).

### 3.3. BSL Assay

After 24 h, all the larvae were alive, swimming, and showing normally visible movements. After 48 h, 68.88% of larvae were active, and 4.44% were in the sublethal stage; the registered mortality was 26.66%. We investigated them under a microscope to observe the changes after 24 and 48 h of exposure. All these microscopic images are presented in Figure 3a–p.

Figure 3 shows that *A salina* larvae are progressively affected by lipid metabolism disturbances, which begin to be visible after 24 h of exposure to F-UBA, compared with blank (Figure 3a–h). Lipid accumulation in the digestive tract and penetration into the neighboring tissues can be observed (Figure 3e–g). In the lower extremity, a slight detachment of the external cuticle from larval tissues is also observed (Figure 3h). After 48 h of exposure, the larval tissues are massively invaded, and the damage continues with digestive blockage and significant tissue destruction (Figure 3m–p) that lead to larvae death. In addition, at the intracellular level, activated lysosomes in cell death processes could be observed using AO staining (Figure 3t–v).

### 3.4. In Vitro Analysis of the Effects of UBA-Loaded Mucoadhesive Oral Films on Human Normal Blood Cells and OSCC CLS-354 Cell Line

The effects of F-UBA on biological processes represented by oxidative stress, caspase 3/7 activity, nuclear shrinkage, autophagy, cell cycle, apoptosis, and DNA synthesis and fragmentation were investigated in blood cell cultures and CLS-354 tumor cell line. All these flow-cytometry analyses follow our preliminary studies about the effects of *U. barbata* dry acetone extract on normal and tumor cells [24,25,53].

#### 3.4.1. ROS Levels

The F-UBA induced cellular oxidative stress in blood cell cultures and CLS-354 tumor cell lines. Total ROS levels were determined through flow cytometry (Figure 4).

In the blood cells treated with F-UBA, ROS levels were substantially higher (543.33 × 10^4^ ± 40.41) than 1% DMSO negative control (242.00 × 10^4^ ± 2.00, *p* < 0.01) and significantly lower than 125 µg/mL of UA positive control: 846.66 × 10^4^ ± 5.77; *p* < 0.01 (Figure 4a,c)

Oxidative stress appreciably increased in the CLS-354 tumor cell line treated with F-UBA (510.00 × 10^4^ ± 10.00) compared with negative control (15.66 × 10^4^ ± 4.04; *p* < 0.01). However, C3UA positive control induced the highest ROS production: 966.66 × 10^4^ ± 57.73, *p* < 0.01 (Figure 4b,d).

#### 3.4.2. Caspase 3/7 Activity

Effector caspase 3/7 enzymatic activity implied in apoptosis was observed after F-UBA treatment for 24 h in normal blood cells and OSCC tumor cells (Figure 5).

In blood cell cultures, after 24 h treatment with F-UBA, we observed significantly decreased values of caspase-3/7 activation mechanisms compared with negative and positive controls: 22.54 ± 1.57 vs. C1: 29.26 ± 1.97, *p* < 0.05; C2UA: 44.74 ± 0.41, *p* < 0.01 (Figure 5a–c,g).

Proapoptotic signal after F-UBA treatment in CLS-354 tumor cell line had significant increasing relative to 1% DMSO negative control and 125 µg/mL UA positive control: 1.79 ± 0.45; vs. 21.88 ± 5.09, *p* < 0.05; 27.02 ± 1.64, *p* < 0.01 (Figure 5d–f,h). 

#### 3.4.3. Nuclear Shrinkage and Autophagy

The presence of pyknotic nuclei (stained with Hoechst 33342) and lysosomal activity (evidenced with AO) in blood cells and CLS-354 tumor cells after 24 h treatment with F-UBA is displayed in Figure 6.

In normal blood cells, F-UBA remarkably diminished the nuclear condensation (4.94 ± 0.39, *p* < 0.01) compared to 1% DMSO negative control (24.50 ± 2.21), as shown in Figure 6a,b,m. However, NS are higher than 125 µg/mL UA positive control (3.19 ± 0.30, *p* < 0.01). The levels of lysosomal activity were considerably lower than the controls: 5.22 ± 0.77; vs. C1: 51.30 ± 3.25; C2UA: 27.05 ± 1.52, *p* < 0.01 (Figure 6g–i,n). 

In CLS-354 tumor cells, after F-UBA treatment, the chromatin shrinkage (NS), and autophagy (A) by Hoechst 33342/AO dual stain, recorded a substantial augmentation compared to C1DMSO negative control: NS: 33.31 ± 2.69 vs. 16.11 ± 3.11, *p* < 0.05; A: 50.07 ± 2.89 vs. 12.57 ± 0.92, *p* < 0.01 (Figure 6d,e,j,k,n). However, the nuclear condensation induced by F-UBA was significantly lower than C2UA positive control: 33.31 ± 2.69 vs. 44.03 ± 0.36; *p* < 0.05 (Figure 6d,f,n). 

#### 3.4.4. Cell Cycle Analysis

Cell cycle analysis was performed by PI/RNase stain (Figure 7) to explore the effects of UBA-loaded mucoadhesive oral films in normal blood cells and CLS-354 tumor cells.

In blood cells (Figure 7a,b,g,i) F-UBA induced a lower cell cycle arrest in G1/G0 phase (85.87 ± 1.30, p < 0.05) and DNA synthesis (1.40 ± 0.65, p < 0.05) than C1DMSO negative control (88.52 ± 0.54, respectively 4.76 ± 0.68). 

In CLS-354 tumor cells, F-UBA treatment reported slowly higher values of cell cycle arrest in G0/G1 than negative control: 93.33 ± 2.42 vs. 92.13 ± 1.61, *p* ≥ 0.05 (Figure 7d,e,h,j). However, DNA synthesis significantly decreased compared to 1% DMSO: 1.05 ± 0.14 vs. 5.47 ± 0.83, *p* < 0.05 (Figure 7d,e,h,j).

#### 3.4.5. Apoptosis

Morphology and cell membrane integrity examination with annexin V-FITC/ PI staining after 24 h treatment with F-UBA in normal blood cells and CLS-354 tumor cells are presented in Figure 8. 

The F-UBA treatment did not induce early cell apoptosis (EA) in normal blood cells compared to C2UA positive control, which determined a considerable level of EA: 37.04 ± 0.66, *p* < 0.01. Therefore, cell viability after F-UBA treatment is significantly higher than 125 µg/mL UA positive control: 94.52 ± 3.85 vs. 61.43 ± 0.88, *p* < 0.01 (Figure 8a,c,g).

The CLS-354 tumor cell viability after F-UBA treatment remained substantially augmented than positive control C2UA: 98.77 ± 1.15 vs. 54.05 ± 1.68, *p* < 0.01 (Figure 8d,f,h). In the CLS-354 tumor cell line, only C2UA determined early apoptosis. EA level is lower than normal blood cells but significantly higher than F-UBA and 1% DMSO (12.92 ± 1.35 vs. 0.00 ± 0.00 *p* < 0.01).

#### 3.4.6. Cell Proliferation

The effects of F-UBA in normal blood cells and CLS-354 tumor cells (DNA synthesis and fragmentation) by EdU incorporation were displayed in Figure 9.

Levels of DNA synthesis, after 24 h of F-UBA treatment in normal blood cells, were considerably diminished than both controls (0.59 ± 0.05 vs. C1: 10.36 ± 1.21; *p* < 0.01; C2UA: 6.49 ± 1.25, *p* < 0.05). Fractional DNA implied in apoptosis, reported significantly higher values compared to positive control: 2.11 ± 0.56 vs. 0.00 ± 0.00, *p* < 0.05 (Figure 9a–c,g,i). 

On the other hand, in CLS-354 tumor cells, UBA-loaded mucoadhesive oral films determined slowly lower levels of DNA synthesis than 1% DMSO negative control (8.55 ± 0.96 vs. C1: 12.44 ± 2.80, *p* ≥ 0.05) and significantly higher than positive control with 125 µg/mL usnic acid (8.55 ± 0.96 vs. 3.14 ± 0.50, *p* < 0.05). However, fractional DNA (subG0/G1) substantially decreased compared to negative control with 1% dimethyl sulfoxide: 4.49 ± 0.87 vs. 15.18 ± 2.17, *p* < 0.05 (Figure 9d–f,h,j). 

#### 3.4.7. Principal Component Analysis

Principal component analysis (PCA) was performed for UBA-loaded mucoadhesive oral patches and both controls (C1-DMSO and C2UA) and variable parameters determined in both cell types (blood cells and CLS-354 tumor cells) according to the correlation matrix and the PCA–correlation circle in the Appendix A.

Two principal components explained the total data variance, with 69.38% attributed to the first (PC1) and 30.62% to the second (PC2). PC1 is associated with controls (C1-DMSO and C2UA), caspase 3/7 activity in both cell types (blood cells and CLS-354 tumor cells), and ROS levels in CLS-354 cells. At the same time, PC2 related to F-UBA mucoadhesive oral films and ROS levels in normal blood cells (Figure 10).

Figure 10 shows that in normal blood cells, caspase 3/7 activation (C3/7) highly positively correlates with early and late apoptosis (*r* = 0.955, *p* > 0.05) and cell cycle arrest in G0/G1 (*r* = 0.930, *p* > 0.05). C3/7 is also moderately correlated with oxidative stress (*r* = 0.681, *p* > 0.05). It shows a strong negative correlation with subG0/G1 (*r* = −0.966, *p* > 0.05) and a moderate one with necrosis (*r* = −0.764, *p* > 0.05). Oxidative stress in normal cells is considerably positively correlated with EA and LA (*r* = 0.867, *p* > 0.05) and negatively correlated with nuclear condensation and subG0/G1 (*r* = −0.900 and −0.848, *p* > 0.05).

In CLS-354 tumor cells, ROS level is substantially positively correlated with nuclear condensation, autophagy, necrosis, and early and late apoptosis (*r* = 0.994, 0.919, 0.886, and 0.854, *p* > 0.05), and negatively correlated with DNA synthesis and subG0/G1 (*r* = −0.993 and −0.951, *p* > 0.05). It shows a moderate negative correlation with cell cycle arrest in G0/G1 (*r* = −0.610, *p* > 0.05) and a minimal positive one with C3/7 (*r* = 0.170, *p* > 0.05). Caspase 3/7 activity is highly negatively correlated with cell cycle arrest in G0/G1 (*r* = −0.884, *p* > 0.05) and moderately positively correlated with necrosis and early and late apoptosis (*r* = 0.608 and 0.658, *p* > 0.05).

Therefore, the placement of F-UBA and both controls (C1-DMSO and C2UA) in the PCA-biplot (Figure 10) was explained, highlighting the corresponding processes triggered in the CLS-354 cancer cell line and normal blood cell cultures.

### 3.5. Antimicrobial Activity

Minimum inhibitory concentrations (MICs) are defined as the lowest concentration of an antimicrobial that will inhibit the visible growth of a microorganism after overnight incubation [71]. According to Phe et al. [72], regarding *S. aureus*, MIC of CTR is 4–8 µg/mL. On *P. aeruginosa*, MIC is substantially higher, varying from 16 µg/mL to over 256 µg/mL (for the highest resistant strains) [73].

The results are displayed in Table 2. 

The first data from Table 2 show the standard antibiotic (CTR), antifungal drug (TRF), and F-UBA microdilutions used. 

The following data from Table 2 show that the colors of standard antibiotics correlate with their inhibiting power and are directly proportional to their concentration. We can observe that CTR microdilutions are over MIC for both tested bacteria. However, S. aureus sensibility at CTR is higher than P. aeruginosa.

The inhibitory activity of UBA-loaded mucoadhesive oral films is higher on *P. aeruginosa* than on *S. aureus.* Therefore, the inhibitory activity of F-UBA of [3.497–0.055] mg/mL against *S. aureus* is similar to CTR of [0.755–0.023] mg/mL. On *P. aeruginosa*, F-UBA of 3.487 mg/mL acts similarly with CTR of 1.603 mg/mL; lower film concentrations of [1.749–0.055] mg/mL have similar effects with CTR of [1.511–0.023] mg/mL.

Data from Table 2 show TRF fungicidal effect on both *Candida* sp. [63] in the entire microdilutions domain.

The F-UBA acts similarly on both *Candida* sp. in a dose-dependent manner. After 24 h incubation with 3.497 mg/mL F-UBA, both *Candida* sp. strains are partially dead [63]. After contact with 1.749 mg/mL F-UBA both fungal strains are alive but do not proliferate. The following F-UBA concentrations [0.874–0.055] mg/mL induced low to moderate proliferation in both *Candida* sp. (Table 2).

## 4. Discussion

In the plant kingdom, lichens are promising sources of antimicrobial and anticancer drugs [74], and *Usnea* sp. is one of the most known representatives as a potent phytomedicine [75]. For almost 6 years, our team studied *U. barbata* from the Călimani mountains. Obtaining various *U. barbata* extracts, quantifying their secondary metabolites, and investigating their pharmacological potential, we aimed to select suitable ones for pharmaceutical formulations. HPLC-DAD analysis of *U. barbata* dry acetone extract (obtained with a yield of 6.36%) shows 282.78 µg/g of usnic acid from a total phenolic content of 862.843 µg/g. Using a complex UHPLC-ESI-OT-MS-MS analysis, Salgado et al. [76] identified other phenolic secondary metabolites which would be extracted in acetone: depsides (e.q., atranorin, lecanoric acid, barbatolic acid) and depsidones (e.q., salazinic acid, norstictic acid). Due to its phenolic constituents, UBA has significant antiradical properties [34,77]. The pharmacological potential of *U. barbata* dry acetone extract was investigated first on *A. salina* larvae as cytotoxicity prescreen and then on the tongue squamous cell carcinoma (CAL-27 cell line) [24,25,77]. Compared to normal fibroblasts (V-79 cell line) [25], UBA cytotoxic effects quantified by MTT assay were higher on CAL-27 tumor cells. The wound healing action was noticeable on tumor cells and minimal on normal cells; the colony formation was substantially inhibited in tumor cells and considerably induced in normal ones. The dry acetone extract acted as a pro-oxidant, causing intense ROS production, and thus stimulating the enzymatic activity of SOD, CAT, and GPx in CAL-27 tumor cells. On V-79 fibroblasts, it acted as an antioxidant, diminishing the levels of antioxidant defense (SOD and CAT) and lipid peroxidation (expressed as MDA levels) [25]. Numerous studies from the scientific literature described in vitro and in vivo studies on liver cells, highlighting the usnic acid hepatotoxicity; therefore, we examined the UBA effects on blood cell cultures due to their various cell types, nucleate white blood cells (WBC), and nucleus-free platelets and red blood cells (RBC) containing neither mitochondria nor nucleus. The *U. barbata* dry acetone extract displayed dose-dependent cytotoxicity; high doses promoted apoptosis and DNA damage in human blood cells by enhancing ROS levels [53]. Only WBC are related to DNA damage.

Moreover, we evaluated the UBA antimicrobial potential on bacterial and fungal strains responsible for opportunistic oral cavity infections in elderly and/or immunocompromised patients [78,79,80]: *S. aureus*, *S. pneumoniae*, *S. pyogenes*, *Enterococcus* sp., other Gram-positive bacteria isolated from oral cavity and pharynx (*S. epidermidis*, *S. oralis*, *S. intermedius*) and *P. aeruginosa* [34,81,82,83]. It was also tested on *C. albicans and C. parapsilosis* [83]. Usnic acid and all phenolic compounds extracted in acetone have antibacterial and antifungal properties and could act synergistically in UBA, and its inhibitory effects are dose-dependent.

All UBA pharmacological activities could have promising applications in oral medicine; therefore, it was incorporated into mucoadhesive oral films. Through physico-chemical and pharmacotechnical analyses, F-UBA properties were compared with reference ones to evaluate their suitability for application to the oral mucosa. A slight weight difference was observed due to the UBA load in the R formulation, and an appropriate uniformity of the films belonging to the same batch was registered, proving that a homogenous drug loading was achieved. The F-UBA thickness allows application on the oral mucosa, being easily acceptable by patients, and its values are included in the optimal range of 0.005–0.200 mm [84]. This property correlates to UBA loaded and is influenced by the type and amount of plasticizer used because the polymer can occasionally increase the film thickness [85]. The folding endurance (expressed as the number of folds needed to break the films or develop visible cracks) is a brittleness indicator [86]. The obtained values over 300 evidence excellent flexibility and hardness due to PEG 400 and HPMC used in the formulations, confirming that UBA incorporation did not modify the film resistance. Pandey and Chauhan [87] developed fast-dissolving sublingual films with Zolmitriptan, and Winarty et al. [88] optimized buccal films with Diltiazem with a similar folding endurance. Tensile strength is the force required to break the oral films [89], while elongation is the film length maximal deformation without damaging it [90]. The mean tensile strength and elongation are also very close between the F-UBA and R, indicating that mucoadhesive oral films exhibit good elastic properties due to HPMC as a matrix-former. The plasticizer increases the flexibility of polymer macromolecules (or macromolecular segments) by loosening the strength of intermolecular tensions [91,92]. Both F-UBA parameters are included in the range of 2.27–4.59 kg/mm^2^ tensile strength and 31.85–54.64% elongation, reported by Kraisit et al. [93] for their buccal films with propranolol nanoparticles, displaying a solid film structure associated with substantial mechanical resistance and flexibility.

The efficiency of a plasticizer agent in mucoadhesive film formulations is primarily determined by the amount used and the polymer–plasticizer interaction. When it comes to an aqueous dispersion, the proportion and amount of partition are influenced by the plasticizer’s solubility in water and its affinity for the polymer. The moisture content affects the mucoadhesive film’s physical stability, preventing them from being brittle. The film has retained water due to the plasticizer hygroscopic attributes, and less than 5% moisture loss is required for better physical strength [94]. As expected, UBA inclusion in the polymer matrix does not considerably change the humidity content of the film. 

The neutral pH confirms that mucoadhesive oral films should not irritate the oral mucosa [95]. In addition, a fast disintegration performance was registered for both formulations, resulting from the intense hydrophilicity of HPMC used as a film-forming polymer. The presence of UBA in the film’s structure delayed the disintegration time by only 8 s compared to the References. The measured swelling rate agreed with the expectations, and no significant difference between both films was assessed. Hashemi et al. [31] optimized oral mucoadhesive patches containing *M. communis* had a disintegration time from 3 h to over 24 h and a swelling ratio between 136–272%.

The presence of hydroxyl groups in the HPMC molecule ensures the matrix stability of the swollen hydrophilic polymer [96]. Soon after swelling starts, the bioadhesion occurs, but the link created is not very strong. Optimal swelling and mucoadhesion only happen at the polymer’s specific hydration level [97,98] and the amount of film-forming polymer and plasticizer strongly influences the retention time over mucosa [99]. It rises with the hydration degree [100], but overhydration causes a rapid decline of mucoadhesive strength due to disentanglement at the polymer/tissue interface. Vasantha et al. [101] developed Eudragit-based mucoadhesive buccal patches of salbutamol sulfate with a mucoadhesion time of 101–110 min.

All pharmacotechnical parameters indicated no substantial differences between F-UBA and R, confirming that UBA-loaded mucoadhesive oral films are suitable for application to the oral mucosa. 

Numerous authors, knowing the bioactivities of plant extracts incorporated in various polymeric films, described only their physico-chemical and pharmacotechnical properties [29,31,102].

In our study, UBA-loaded mucoadhesive oral films have neutral pH, and fast disintegration permitted pharmacological potential evaluation through in vivo and in vitro studies.

The F-UBA investigation was performed using similar studies evaluating *U. barbata* dry acetone extract. Thus, the cytotoxicity prescreen was performed on A. salina larvae. Anticancer activity was assessed on a similar OSCC cell line (CLS-354, mouth epithelial squamous cell carcinoma). Blood cell cultures were used as normal cells because buccal mucosa has a substantial blood supply and is permeable for most blood cells [103]. The two structural layers of the oral mucosa are the outer epithelium and the underlying lamina propria (LP). The buccal epithelium is a physical and chemical barrier, but WBCs pass through the epithelium to LP, constituting the immunological defense of the oral cavity. Conventionally, resident cells in the oral mucosa are the ones of stromal origin, such as gingival keratinocytes, fibroblasts, and periodontal ligament cells [104]. Migratory cells are lymphocytes (with T cells) and segmented cells (polymorphonuclear, including neutrophils and eosinophils). In oral cancer, the lymphocytes number decreases [105]. 

In addition, various substances go through the oral mucosa by simple diffusion, reaching the capillary area under normal conditions. The UBA-loaded mucoadhesive films are projected to remain on oral mucosa different lesions for 85 min. Therefore, it is essential to know their effects on blood cells because sensitization and bleeding are much more frequent in the buccal mucosa. The blood samples were collected from a single donor (the same as previously described in UBA biological studies).

The UBA has a considerable UA content, and F-UBA action is expected to be similar to that of the positive control (C2UA). Data analysis shows that F-UBA selectively affected tumor cells. Compared with C1DMSO negative control, it increased ROS levels, nuclear condensation, autophagy, and cell cycle arrest in G0/G1 and diminished DNA synthesis in the CLS-354 (oral epithelial squamous cell carcinoma) tumor cell line. Inducing high oxidative stress, F-UBA triggered the previously mentioned processes that lead to CLS-354 cancer cells’ death. The F-UBA exhibited a protective effect on normal blood cells, diminishing the apoptotic processes due to 1% DMSO: caspase 3/7 activation, cell cycle arrest in G0/G1, nuclear condensation, and autophagy.

Usnic acid, the main bioactive metabolite in the *Usnea* lichens, is soluble in acetone and is found in a significant amount in the *U. barbata* dry acetone extract [34]. Assessing the effects of UA (100–300) mg/kg on adult male rats, Alahmadi et al. [106] reported that hepatocytes had an increased lipid droplet, swollen mitochondria, and fragmented rough endoplasmic reticulum and cell membrane damage. These harmful effects induced by UA on the cellular level are similar to the present study’s results on *A. salina* larvae, used as an animal model. After 24 h of exposure to F-UBA, the lipids accumulation in the digestive tube and their progressive penetration in neighboring tissues lead to massive cellular damage and brine shrimp larvae death after 48 h.

Nevertheless, UA exhibits beneficial bioactivities (antimicrobial, anticancer, antioxidant), displaying a dual redox behavior (pro-oxidant in tumor cells and antioxidant, protective, in normal cells [107], and the present results confirmed it. Andania et al. also proved the usnic acid antitumor effect on the OSCC cell line (HSC-3). In *Usnea* sp. extracts, the primary quantified metabolite is usnic acid; the others are not commonly determined individually and are quantified as total phenolic content. The dry acetone extract loaded in F-UBA contains total phenolic compounds of 862.84 mg PyE/g, from which 241.83 mg/g is usnic acid [34]. The action of F-UBA on CLS-354 tumor cells through high cellular oxidative stress was similar to usnic acid of 125 mg/mL; it had a lower intensity due to the significant difference in UA content. The usnic acid phenolic structure is responsible for all bioactivities at the cellular level [108,109,110].

The antimicrobial activity of oral films containing plant extracts is evaluated through various methods. For example, Gajdziok et al. [111] applied a microbial suspension (*S. aureus* and *C. albicans*) on the surface of their mucoadhesive oral patches. After incubation, the patches were examined by direct imprinting on a solid culture medium; the growth of bacteria and yeast strains was observed and expressed as the number of colony-forming units (CFU) on the surface of the imprint. Chiaoprakobkij et al. [33] performed antibacterial studies on *S. aureus* and *E. coli*, counting the living bacterial cells on the films loaded with *Garcinia mangostana* L. extract. Other authors also selected *S. aureus* and *E. coli* for antibacterial activity evaluation of their poly(vinyl alcohol)/plant extracts films [112]. They measured the inhibition zone diameter (IZD), and the interpretation of the results used Poly (vinil alcohol) as a reference. It had no inhibitory activity (IZD = 0 cm) and was noted with (−)—no action. The highest values were registered as (++++ and +++)—Very good activity, followed, in decreasing order, by (++)—good activity and (+)—weak activity. The appreciation of their inhibitory activity was the most part, proving in qualitative mode this property of the films with plant extracts without using a standard antibiotic drug [112].

In a great measure, these qualificatives are similar to the resazurin dye chart for a rapid evaluation of antibacterial [61] and antifungal [63] activities. Our study examined the F-UBA inhibitory effects against the most common bacterial and fungal species implicated in oral infections in immunocompromised patients.

Directly applied to the oral malignant lesion, the UBA-loaded mucoadhesive oral films could exert a topical anticancer effect, reducing the progression of oral squamous cell carcinoma. This formulation could also help through other mechanisms [113]. Because OSCC decreases immune defense, the oral cavity becomes more susceptible to bacterial and fungal infections. Therefore, UBA-loaded mucoadhesive oral films could inhibit *S. aureus*, *P. aeruginosa*, and *Candida* sp., which are responsible for opportunistic oral infectious diseases in immunocompromised patients [78,79]. These results proved the chemotherapeutic effect of UBA-loaded mucoadhesive oral films. F-UBA could also display a chemopreventive action. Due to their cytoprotective effect and inhibitory activity against *C. albicans*, F-UBA can maintain a low risk because this fungal species is implied in OSCC generation, according to Prakash et al. [14]. The third condition, sensitizing in multidrug resistance, according to Dehelean et al. [13], could be supported by usnic acid’s powerful effect-enhancing and toxicity-reducing in ascitic tumor-bearing mice treated with bleomycin [114]; usnic acid inhibits angiogenesis in vascular endothelial growth factor (VEGF) model and chick embryo [115]. Dar et al. [82] consider combination therapy in cancer essential for declining drug resistance, reducing tumor development, blocking mitotic cells, and inducing apoptosis. At the same time, the toxicity of monotherapy could be minimized. The results suggest that UBA-loaded in mucoadhesive oral films could be considered a combination in oral squamous cell carcinoma therapy due to the synergic action of usnic acid and other secondary phenolic metabolites.

None of the numerous analyzed mucoadhesive film compositions included lichen extracts, even though lichens are known for their antitumor, antibacterial, antifungal, antiviral, and enzyme inhibitory properties. Recognized in Traditional Chinese Medicine (TCM) as “Song Lo,” *Usnea* lichens were consumed as tea or decoction for liver detoxification [116]. The typical TCM dosages are 6–9 g of dried lichen, corresponding to approximately 60–120 mg of usnic acid per day. Usnic acid was included in fat-burning products such as UCP-1 (BDC Nutrition, Richmond, Kentucky, which contains usnic acid, L-carnitine, and calcium pyruvate) and Lipo Kinetix (Syntrax Innovations Inc., Cape Girardeau, Missouri, containing norephedrine hydrochloride, sodium usniate, 3,5-diiodothyronine, yohimbine hydrochloride, and caffeine) [117] and associated with severe hepatotoxicity. In the present study, each F-UBA evaluated for potential therapeutic application contains 175 µg UBA, corresponding to a usnic acid content of 42.32 µg, a value more than 1400 times lower than the minimal dose allowed by TCP.

## 5. Conclusions

In this study, mucoadhesive oral films containing *U. barbata* dry acetone extract were manufactured using HPMC K100 as a polymer matrix and PEG 400 as a plasticizer. Complex physico-chemical and pharmacotechnical analyses proved their suitability for oral administration.

The pharmacological evaluation confirmed F-UBA in vitro anticancer activity on oral squamous cell carcinoma based on high oxidative stress induced in CLS-354 cells. The results also revealed F-UBA cytoprotective action on normal cells and the dose-dependent growth inhibition of bacterial and fungal pathogens involved in immunosuppressed patients’ oral infections.

Therefore, the present study suggests that UBA-loaded mucoadhesive oral films could be a helpful phytotherapeutic formulation in the complementary treatment of oral squamous cell carcinoma. Further in vivo and clinical research could be following steps in the F-UBA analysis, aiming to confirm their medical benefits.

## Figures and Tables

**Figure 1 antioxidants-11-01934-f001:**
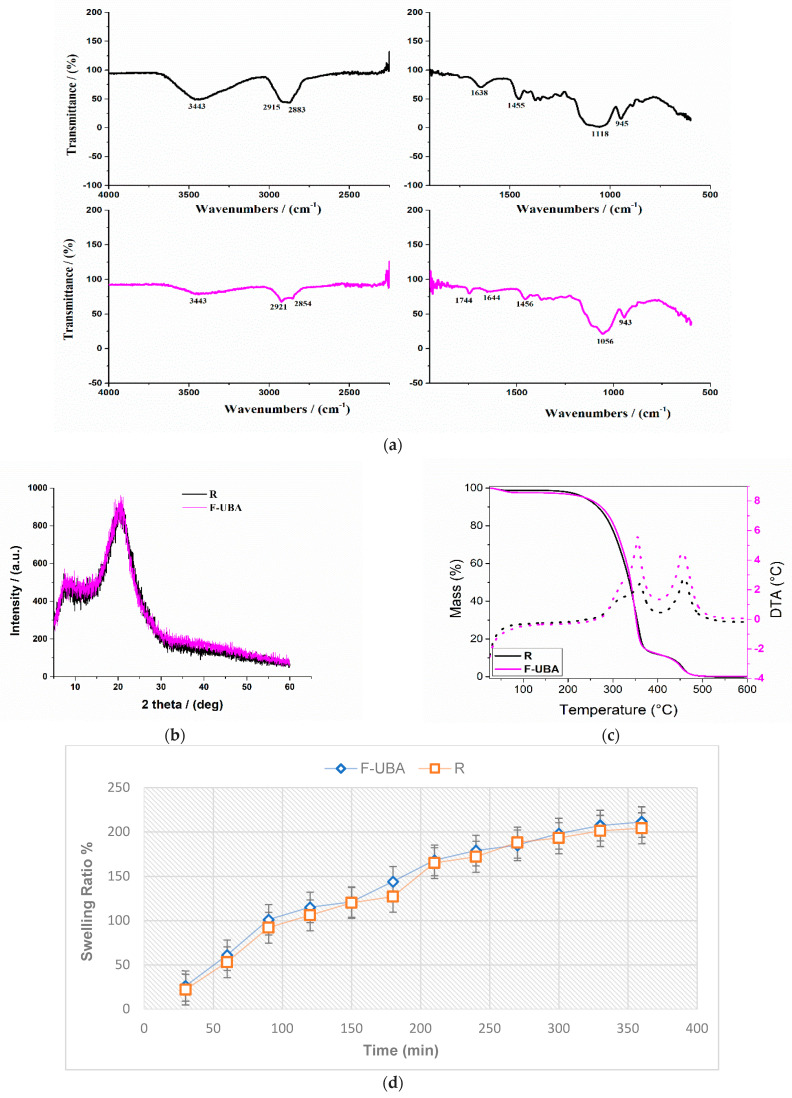
(**a**) FTIR spectra of mucoadhesive oral films R (black line) and F-UBA (magenta line); (**b**) XRD pattern and (**c**) TG/DTA curves; (**d**) swelling ratio over 6 h of both mucoadhesive films (F-UBA and R). F-UBA—UBA-loaded mucoadhesive oral film; R—Reference (mucoadhesive oral film without UBA); UBA—*U. barbata* dry acetone extract.

**Figure 2 antioxidants-11-01934-f002:**
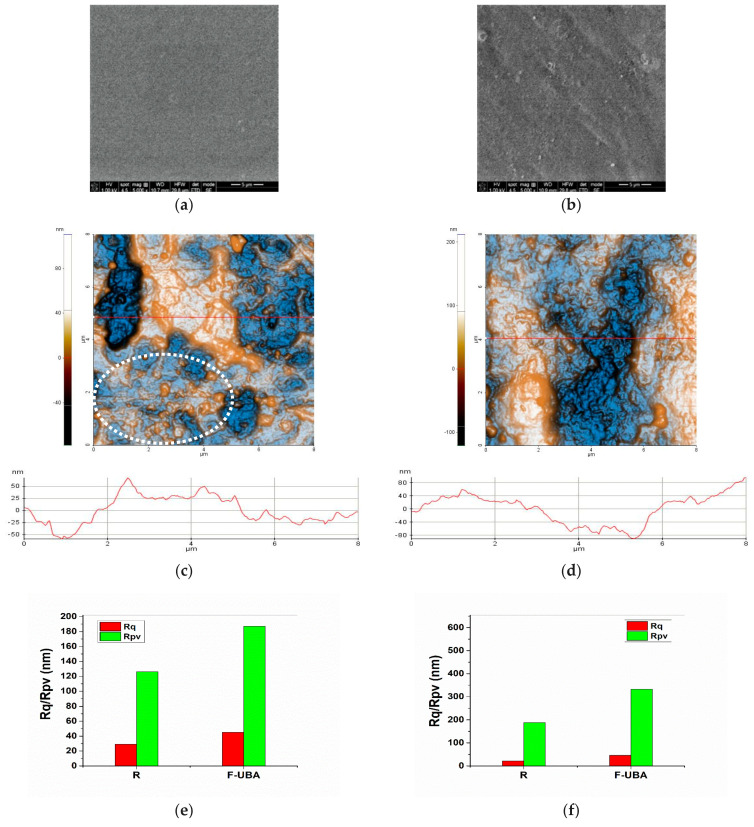
SEM and AFM images of mucoadhesive oral films. SEM micrographs (**a**,**b**) of both mucoadhesive oral films: (**a**) reference (R); (**b**) F-UBA. 2D AFM images (**c**,**d**) in enhanced contrast view at the scale of (8 × 8) µm^2^ together with representative line scans for films: (**c**) R, (**d**) F-UBA; Roughness (Rq), and peak-to-valley (Rpv) parameters (**e**) along the line-scans and (**f**) for the whole scanned areas; and at the scale of (3 × 3) µm^2^ together with representative line-scans for films (**g**) R and (**h**) F-UBA. R—Reference (mucoadhesive oral film without UBA); F-UBA—UBA-loaded mucoadhesive oral film; UBA—*U. barbata* dry acetone extract.

**Figure 3 antioxidants-11-01934-f003:**
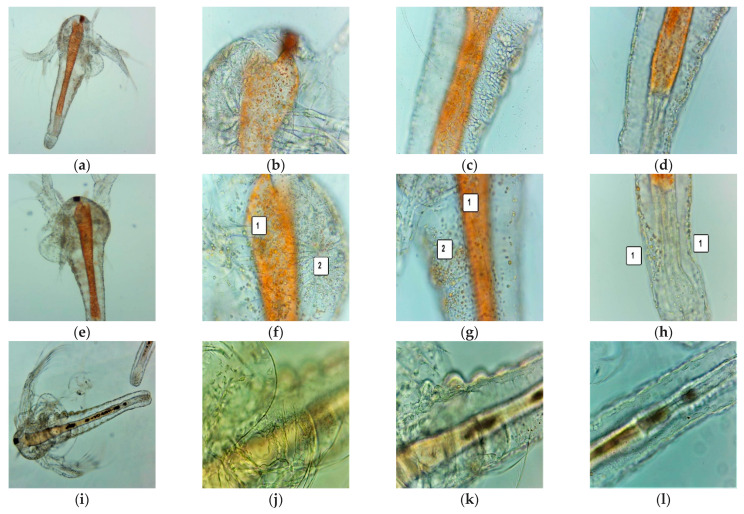
(**a**–**p**) *A. salina* larvae after the exposure period of 24 and 48 h to F-UBA—microscopic images at 100× (**a**,**e**,**i**,**m**) and 400× (**b**–**d**,**f**–**h**,**j**–**l**,**n**–**p**). After 24 h: blank (**a**–**d**) and F-UBA (**e**–**g**); after 48 h: blank (**i**–**l**) and F-UBA (**m**–**p**). The following changes after 24 h can be observed compared with blank: (**f**–**g**) lipids accumulation in the digestive tract (1), lipids penetration into the neighboring tissues (2); (**h**) a low detachment of the cuticle from larval tissues (1); (**n**) cell damage with large intercellular spaces and tissue destruction (1) and considerable detachment of the cuticle from larval tissues (2); (**o**) detachment of the membrane that separates the digestive tube from the neighboring tissues (1), lipid penetration in the adjacent tissues (2), and considerable detachment of the external cuticle from larval tissues (3); (**p**) digestive blockage (1) with massive tissue destruction (2) and high detachment of the external cuticle from larval tissues (3). (**q**–**v**) FM images of *A. salina* larvae after 48 h of exposure at F-UBA—stained with Acridine Orange 400× (**q**–**s**) and 200× (**t**–**v**); (**q-s**)—blank; (**t**–**v**)—F-UBA. The red fluorescence shows intracellular lysosomes activated in cell death processes (**t**–**v**). F-UBA—UBA-loaded mucoadhesive oral film; UBA—*U. barbata* dry acetone extract.

**Figure 4 antioxidants-11-01934-f004:**
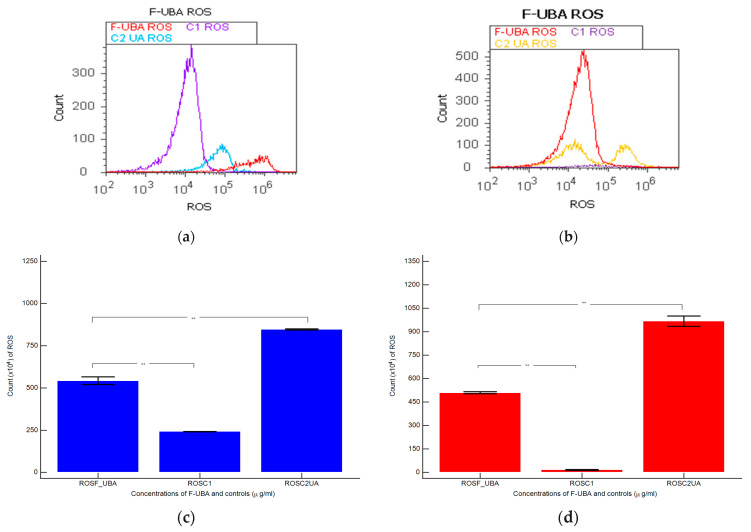
Reactive oxygen species (ROS) in blood cells (**a**) and CLS-354 tumor cells (**b**) after 24 h treatment with F-UBA; C1—1% DMSO negative control; C2—125 µg/mL UA positive control; Statistical analysis of reactive oxygen species (ROS) in blood cell cultures (**c**) and CLS-354 tumor cell line (**d**), after 24 h treatment with F-UBA; *** p* < 0.01 represents significant statistical differences between controls and sample made by paired samples *t*-test. C1—negative control with 1% dimethyl sulfoxide (DMSO); C2UA—positive control with 125 µg/mL usnic acid (UA); F-UBA—UBA-loaded mucoadhesive oral film; UBA—*U. barbata* dry acetone extract.

**Figure 5 antioxidants-11-01934-f005:**
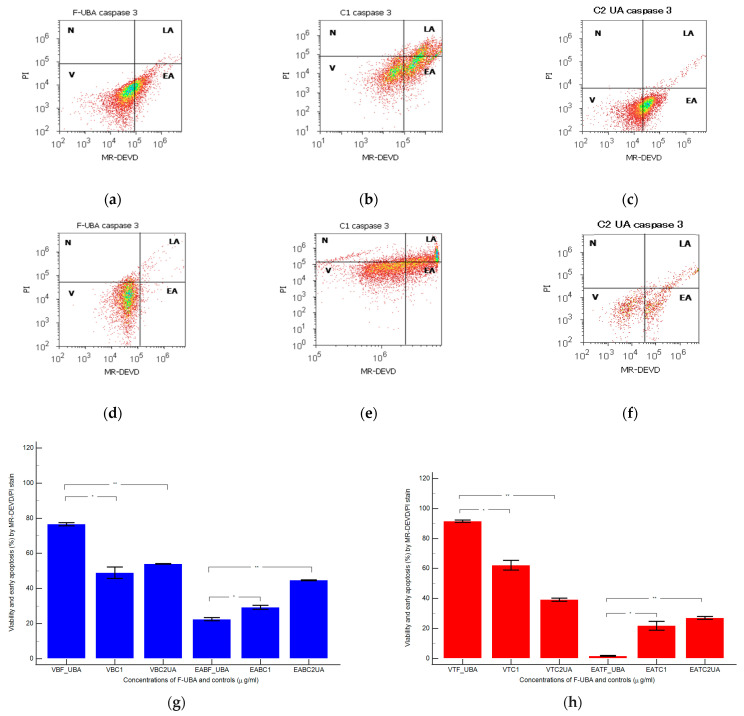
Activity of 3/7 effector caspases in blood cells (**a**–**c**,**g**) and CLS-354 tumor cells (**d–f**,**h**) after 24 h treatment with F-UBA. MR-DEVD/PI patterns of F-UBA (**a**,**d**), 1% DMSO negative control (**b**,**e**); 125 µg/mL UA positive control (**c**,**f**). Statistical analysis of 3/7 caspases activity in blood cells (**g**) and CLS-354 tumor cells (**h**) after 24 h treatment with F-UBA; * *p* < 0.05 and ** *p* < 0.01 represent significant statistical differences between controls and film made by paired samples *t*-test. V—viability; EA—early apoptosis; C1—negative control with 1% dimethyl sulfoxide (DMSO); C2UA—positive control with 125 µg/mL usnic acid (UA); F-UBA—UBA-loaded mucoadhesive oral film; UBA—*U. barbata* dry acetone extract.

**Figure 6 antioxidants-11-01934-f006:**
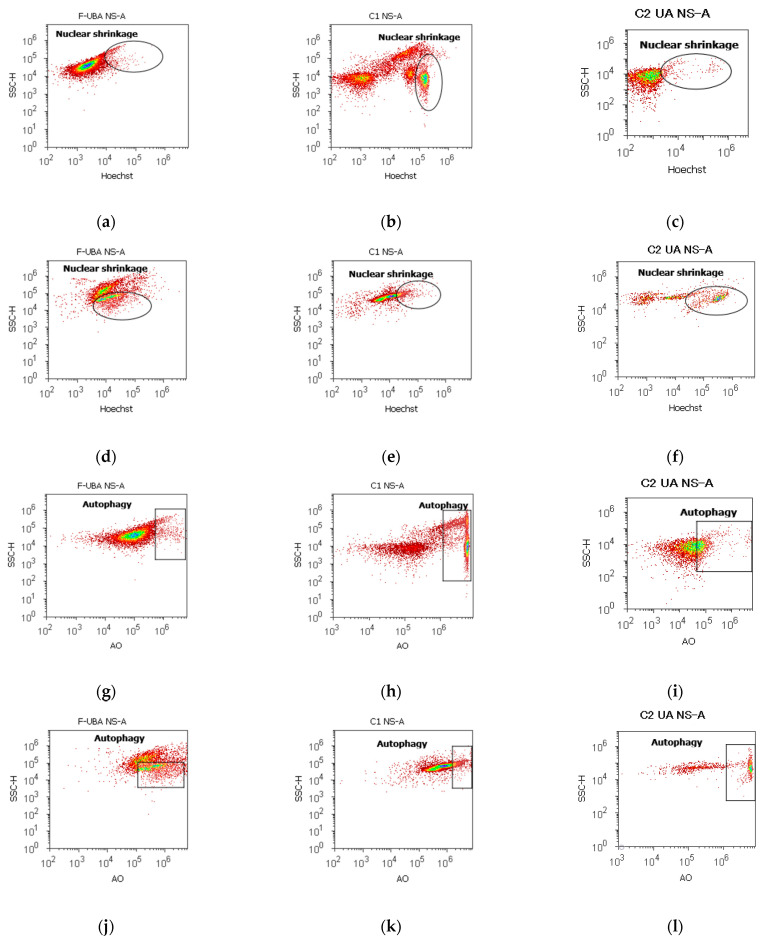
Nuclear shrinkage (**a**–**f**) and lysosomal activity (**g**–**l**) in blood cells (**a**–**c**,**g**–**i**) and CLS-354 tumor cells (**d**–**f**,**j**–**l**) after 24 h treatment with F-UBA. Hoechst patterns of F-UBA (**a**,**d**); acridine range patterns of F-UBA (**g**,**j**); 1% DMSO negative control (**b**,**e**,**h**,**k**); 125 µg/mL UA positive control (**c**,**f**,**i**,**l**). F-UBA—UBA-loaded mucoadhesive oral film; UBA—*U. barbata* dry acetone extract. Statistical analysis of nuclear shrinkage and autophagy in normal blood cells (**m**) and CLS-354 tumor cells (**n**) after 24 h treatment with F-UBA; * *p* < 0.05 and ** *p* < 0.01 represent significant statistical differences between controls and sample made by paired samples *t*-test. NS—nuclear shrinkage; A—autophagy; C1—negative control with 1% dimethyl sulfoxide (DMSO); C2UA—positive control with 125 µg/mL usnic acid (UA); F-UBA—UBA-loaded mucoadhesive oral film; UBA—*U. barbata* dry acetone extract.

**Figure 7 antioxidants-11-01934-f007:**
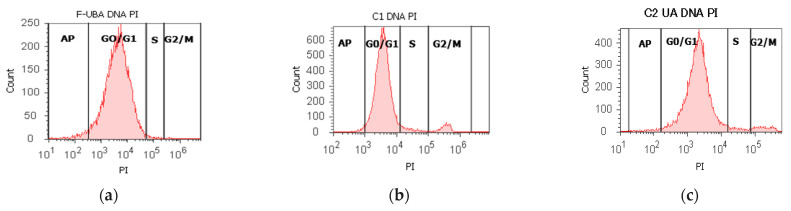
Cell cycle analysis in normal blood cells (**a**–**c**) and CLS-354 tumor cells (**d**–**f**) after 24 h treatment with F-UBA; PI/RNase patterns of F-UBA (**a**,**d**); 1% DMSO negative control (**b**,**e**); 125 µg/mL UA positive control (**c**,**f**); F-UBA and both controls extrapolated on PI axis (**g**,**h**). AP—apoptosis (subG0/G1); PI—propidium iodide; S—synthesis of cell cycle phases. Statistical analysis of G0/G1, synthesis (S), and G2/M phases of cell cycle in normal blood cell cultures (**i**) and CLS-354 tumor cell line (**j**) after 24 h treatment with F-UBA; * *p* < 0.05 and ** *p* < 0.01 represent significant statistical differences between controls and sample made by paired samples *t*-test. C1—negative control with 1% dimethyl sulfoxide (DMSO); C2UA—positive control with 125 µg/mL usnic acid (UA); F-UBA—UBA-loaded mucoadhesive oral film; UBA—*U. barbata* dry acetone extract.

**Figure 8 antioxidants-11-01934-f008:**
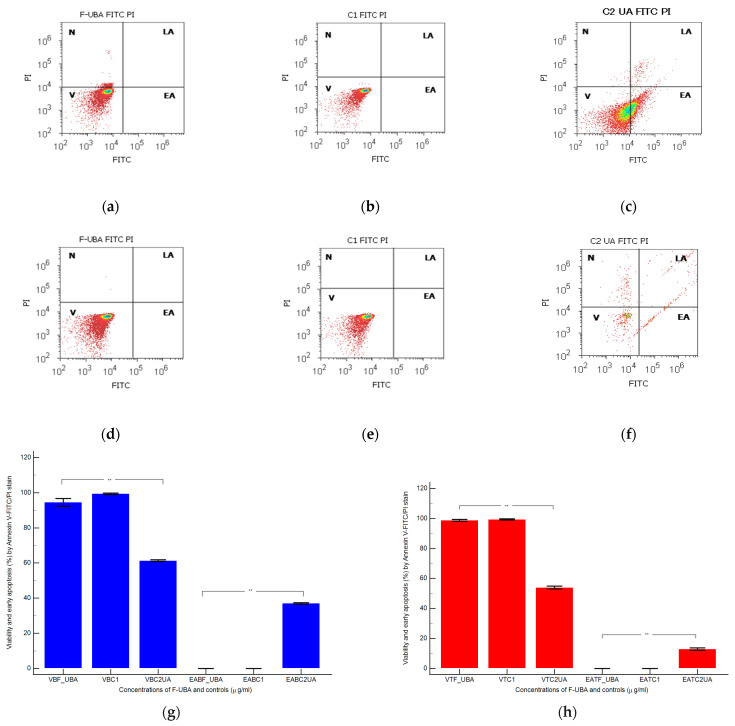
Cell apoptosis in normal blood cells (**a**–**c**) and CLS-354 tumor cells (**d**–**f**) after 24 h treatment with F-UBA; Annexin V-FITC/PI patterns of F-UBA (**a**,**d**); 1% DMSO negative control (**b**,**e**); 125 µg/mL UA positive control (**c**,**f**). Statistical analysis of cell apoptosis in normal blood cell cultures (**g**) and CLS-354 tumor cell line (**h**) after 24 h treatment with F-UBA; ** *p* < 0.01 represents significant statistical differences between controls and sample made by paired samples *t*-test. V—viability; EA—early apoptosis; C1—negative control with 1% dimethyl sulfoxide (DMSO); C2UA—positive control with 125 µg/mL usnic acid (UA); F-UBA—UBA-loaded mucoadhesive oral film; UBA—*U. barbata* dry acetone extract.

**Figure 9 antioxidants-11-01934-f009:**
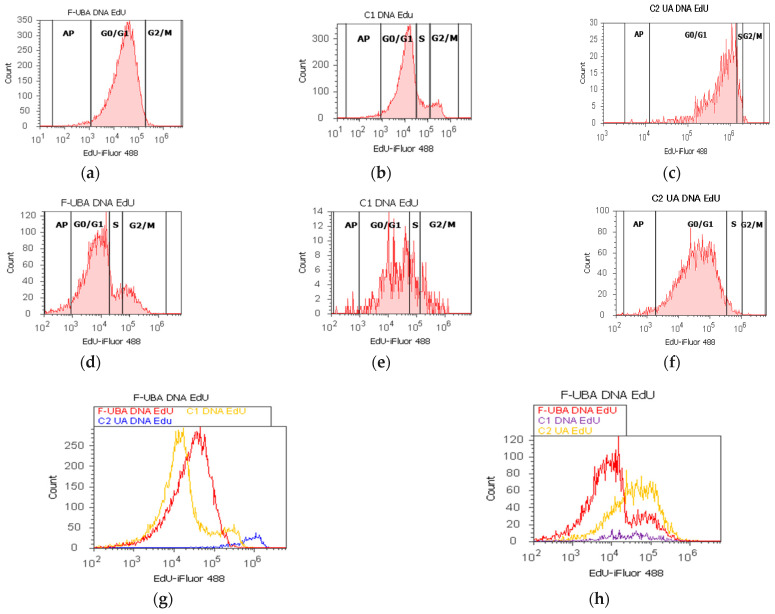
Synthesis (S) and fragmentation of DNA (subG0/G1) in normal blood cells (**a**–**c**) and CLS-354 tumor cells (**d**–**f**) after 24 h treatment with F-UBA; EdU-iFluor 488 patterns of F-UBA (**a**,**d**); 1% DMSO negative control (**b**,**e**); 125 µg/mL UA positive control (**c**,**f**). F-UBA and controls extrapolated on EdU-iFluor 488 axis (**g**,**h**). AP—apoptosis (subG0/G1). Statistical analysis of cell proliferation in blood cell cultures (**i**) and CLS-354 tumor cell line (**j**) after 24 h treatment with F-UBA; * *p* < 0.05 and *** p* < 0.01 represent significant statistical differences between controls and sample made by paired samples *t*-test. C1—negative control with 1% dimethyl sulfoxide (DMSO); C2UA—positive control with 125 µg/mL usnic acid (UA); F-UBA—UBA-loaded mucoadhesive oral film; UBA—*U. barbata* dry acetone extract.

**Figure 10 antioxidants-11-01934-f010:**
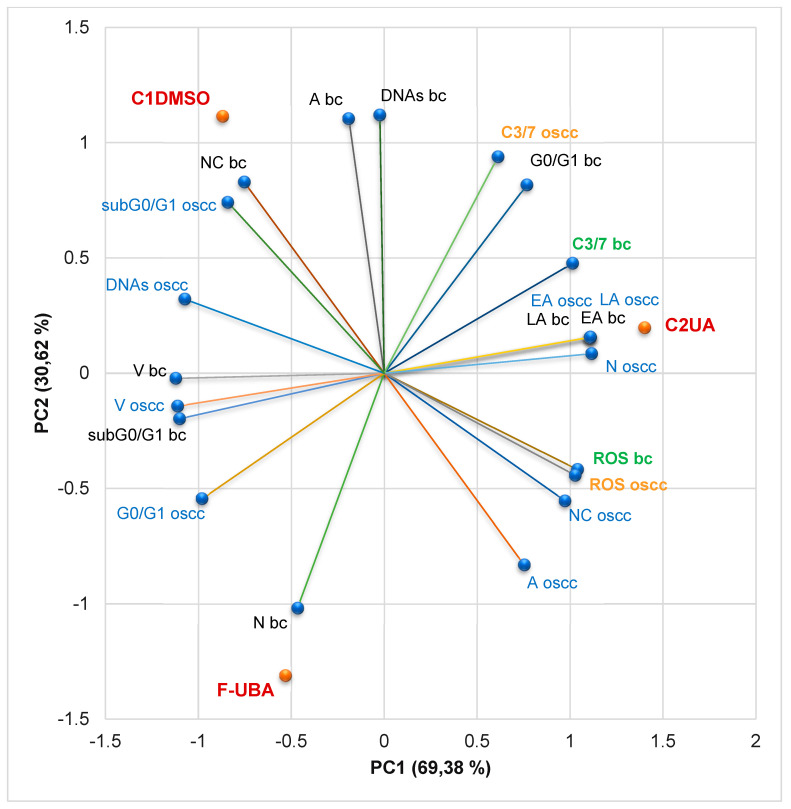
PCA-Correlation biplot between mechanisms (caspase 3/7 activity and cellular oxidative stress) and processes induced by F-UBA and both controls (C1-DMSO and C2UA) in blood cells (bc) and CLS-354 tumor cells (oscc). V—viability, EA—early apoptosis, LA—late apoptosis, N—necrosis, NC—nuclear condensation, A—autophagy, DNAs—DNA synthesis, G0/G1—cell cycle arrest in G0/G1, ROS—oxidative stress, C3/7—caspase 3/7 activity, F-UBA—UBA-loaded mucoadhesive oral films.

**Table 1 antioxidants-11-01934-t001:** Composition, physico-chemical properties, and pharmacotechnical characterization of mucoadhesive oral films (F-UBA and R).

Variable		F-UBA	R
*Composition*			
UBA (g)		0.25	-
Ethyl alcohol 96% (*v*/*v*) (g)		5	5
Isopropyl alcohol (g)		5	5
PEG 400 (g)		5	5
HPMC 15% water dispersion (*w*/*w*) (g)		84.75	85
*Physico-chemical properties-TG/DTA parameters*			
1st Step (%)	TG (%)	2.50%	1.20%
2nd Step (%)	TG (%)/Tmax (°C)	85.5%/355.2 °C	86.9%/360.5 °C
3rd Step (%)	TG (%)/Tmax (°C)	12.1%/461.8 °C	11.9%/454.7 °C
*Pharmacotechnical properties*			
Weight uniformity (mg)		70 ± 3.54	66 ± 4.18
Thickness (mm)		0.060 ± 0.002	0.058 ± 0.003
Folding endurance value		>300	>300
Tensile strength (kg/mm^2^)		3.02 ± 2.39	2.88 ± 3.43
Elongation %		47.26 ± 2.16	49.25 ± 2.24
Moisture content % (*w*/*w*)		4.11 ± 0.35	3.98 ± 1.02
pH		7.01 ± 0.01	7.04 ± 0.02
Disintegration time (seconds)		146 ± 5.09	138 ± 4.67
Swelling ratio (% after 6 h)		211 ± 4.31	204 ± 3.29
Ex vivo bioadhesion time (minutes)		85 ± 2.33	82 ± 2.61

F-UBA—mucoadhesive oral films loaded with *U. barbata* dry acetone extract; R—Reference (films without UBA); UBA—*U. barbata* dry acetone extract; PEG 400—polyethylene glycol 400; HPMC—hydroxypropyl methylcellulose.

**Table 2 antioxidants-11-01934-t002:** Initial concentrations and microdilutions of standard antibacterial and antifungal drugs and sample (F-UBA) and antimicrobial activity of UBA-loaded mucoadhesive oral films.

Micro-Dilution	CTR (mg/mL)	TRF (mg/mL)	F-UBA (mg/mL)
30.230 ± 0.630	122.330 ± 0.850	10.050 ± 0.180	70 ± 3.540
1	1.511 ± 0.043	6.117 ± 0.042	0.500 ± 0.009	3.497 ± 0.172
2	0.755 ± 0.022	4.893 ± 0.034	0.250 ± 0.004	1.749 ± 0.086
3	0.377 ± 0.011	3.914 ± 0.027	0.125 ± 0.002	0.874 ± 0.043
4	0.188 ± 0.005	3.131 ± 0.021	0.061 ± 0.001	0.438 ± 0.022
5	0.094 ± 0.002	2.505 ± 0.017	0.031 ± 0.001	0.219 ± 0.011
6	0.047 ± 0.002	2.004 ± 0.014	0.015 ± 0.001	0.110 ± 0.006
7	0.023 ± 0.001	1.603 ± 0.011	0.007 ± 0.001	0.055 ± 0.003
*S. aureus*	*P. aeruginosa*
**F-UBA**	**CTR**	**F-UBA**	**CTR**
**A ***	**B ****	**A ***	**B ****	**A ***	**B ****	**A ***	**B ****
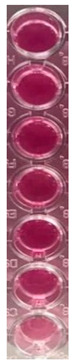	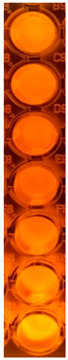		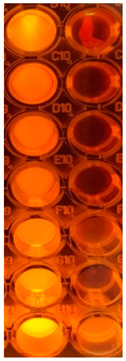	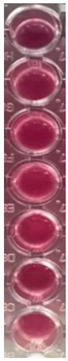	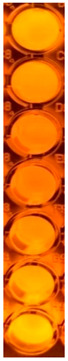		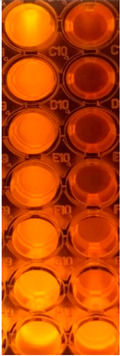
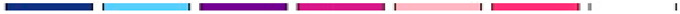 *
*C. albicans*	*C. parapsilosis*	Color ***	Score ***	Signification ***
TRF	F-UBA	TRF	F-UBA
	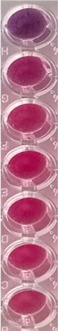		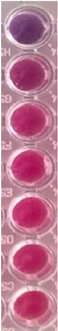	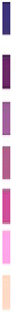	0	Blue—cells are dead
1	Violet-blue—cells are partially dead
2	Violet—cells are alive; no proliferation
3	Light-violet—low proliferation
4	Dark pink—moderate proliferation
5	Pink—fast proliferation
6	Light pink—very fast proliferation

* Interpretation of obtained results adapted from Madushan et al. [61] (A); blue: excellent; light blue—very good, violet—good, purple pink—moderate, light pink—low, pink—very low, white—no effect; CTR—Ceftriaxone, F-UBA—Mucoadhesive film with *U. barbata* dry extract in acetone. ** Well plates examined at a wavelength of 470 nm (B). *** Results interpreting adapted from Bitacura et al. [63]. TRF—Terbinafine.

## Data Availability

Data are contained within the manuscript.

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
