# Peer review of "In Vitro Anticancer Activity of Mucoadhesive Oral Films Loaded with Usnea barbata (L.) F. H. Wigg Dry Acetone Extract, with Potential Applications in Oral Squamous Cell Carcinoma Complementary Therapy"

_antioxidants, 2022, doi:10.3390/antiox11101934_

Round 1

Reviewer 1 Report

The article entitled: “In vitro Anticancer Activity of Mucoadhesive Oral Films loaded with Usnea barbata (L.) F. H. Wigg Dry Acetone Extract, for Potential Applications in Oral Squamous Cell Carcinoma’s Complementary Therapy” studies the possibility of UBA-loaded mucoadhesive oral films for potential complementary treatment of OSCC.

The interest of the scientific community in the use of herbal and/or alternative medicines for different ailments, is growing with the years. For this reason, the study presented shares a merit of interest.

In my opinion, the major flaw of the study is the comparison of the OSCC cell line to human blood derived from healthy donor. Why is the cell line not compared to a non-malignant cell line from the same area (oral cavity) from humans?

If there is an explanation to this, it must be reported to the manuscript.

Other comments regard

The introduction:

does it need to be starting from what is cancer? Lines from 76 to 84 are unnecessary. The authors could start introducing the prevalence of oral cancer and why it is important to research possible treatments.

Lines 111+: The principal mechanisms triggered in OSCC tumor cell line are apoptosis, DNA damage, oxidative stress, and cell cycle arrest [11,12,18,19]. A coherent order with the previous sentence is needed.

The last paragraph of the introduction is about managing oral submucous fibrosis. Since the introduction to the research subject is already lengthy, I suggest that this paragraph can be omitted.

The Results

Line 473, 945 in the figure

Line 643, the text refers to 7g,h but there is no g and h in the Figure. The legend goes up to (f).

Line 645, the red fluorescence, is not so obvious to the figures.

General comments

Line 46, there is a parenthesis, which is not placed there correctly

Line 132, and 270 in vitro, needs to be in italics.

Line 281, ex vivo, needs to be in italics.

Lines 350 and 352, repetition of the word “Then”

Line 366, the sentence should not start with a number.

Line 846, 24 hours instead of 24 ore

Table 5, cells are dead instead of cells are death

Line 873, Nam et al. (2017), has different reference appearance from the rest of the text. There should be uniformity.

Line 916, could a sentence start with an abbreviation?

Reviewer 2 Report

Although the article is very interesting I would like to point out a few important things.

I wonder why authors hav  choosen Antioxidants. Even the titlle did not reflect the scope of journal.

Morover among over a dozen different tests only several procent concerns antioxidants (ROS levels). "Antioxidant" aspects of research is at very weak level dicussed with other results of numerous tests.

This article would perfectly fit for Molecules or another journal's scope.

Discussion is feeble, does not correspond with other papers, and contains useless fragments like lines 891-896 and numerous repetitions.

Concerning antimicrobial activity. How authors have choosen the microorganism for the test? Why only four (two bacteria and two Candida species)? Actually, in what units the antimicrobial activity is expressed? What were the criteria to qualify this kind of activity? If activity at 122 mg/mL is good, the 30 mg/mL should be moderate or very good. Nevertheless is concerned to be rather as weak.

Round 2

Reviewer 1 Report

The authors have answered all comments adequately and I recommend the publication of the manuscript on its final version.

Reviewer 2 Report

Correction accepted.